# Morphological Study on the Differentiation of Flower Buds and the Embryological Stages of Male and Female Floral Organs in *Lespedeza davurica* (Laxm.) Schindl. cv. JinNong (*Fabaceae*)

**DOI:** 10.3390/plants13121661

**Published:** 2024-06-15

**Authors:** Lirong Tong, Juan Wang

**Affiliations:** 1College of Grassland Science, Shanxi Agricultural University, Taigu 030800, China; maria-dawn@cau.edu.cn; 2Forage Seed Laboratory, College of Grassland Science and Technology, China Agricultural University, Beijing 100193, China

**Keywords:** *Lespedeza davurica*, *Fabaceae*, stamens, pistils, double fertilization

## Abstract

*Lespedeza davurica* (Laxm.) is a leguminous plant with significant ecological benefits, but its embryonic development mechanism remains unclear. We investigated the flower bud differentiation, megaspore and microspore formation, gametophyte development, and embryo and endosperm development in *L. davurica*. Our aim was to elucidate the relationship between the external morphology and internal development processes of male and female floral organs during growth, as well as the reproductive factors influencing fruiting. The results indicated that although the pistil develops later than the stamen during flower bud differentiation, both organs mature synchronously before flowering. *L. davurica* pollen exhibits three germination grooves, a reticulate outer wall, and papillary structures on the anther surface. In vivo pollination experiments revealed abnormal spiral growth of *L. davurica* pollen tubes within the style and the occurrence of callus plugs, which may reduce the seed setting rate. The anther wall development follows the dicotyledonous type, with tetrads formed through microspore meiosis exhibiting both left–right symmetry and tetrahedral arrangements. *L. davurica* has a single ovule, and the embryo sac develops in the monosporic polygonum type. After dormancy, the zygote undergoes multiple divisions, progressing through spherical, heart-shaped, and torpedo-shaped embryo stages, culminating in a mature embryo. A mature seed comprises cotyledons, hypocotyl, embryo, radicle, and seed coat. Phylogenetic tree analysis reveals a close genetic relationship between *L. davurica* and other leguminous plants from the genera *Lespedeza* and *Medicago*. This study provides valuable insights into the regulation of flowering and hybrid breeding in leguminous plants and offers a new perspective on the development of floral organs and seed setting rates.

## 1. Introduction

Plant development depends on the activity of various meristems that produce organs such as leaves and floral organs throughout the life cycle. Every step of plant reproductive development is affected by multiple factors; any defects in this process may lead to failure of the reproductive process, thus threatening the survival of the species [1]. Flower bud differentiation and megaspore and microspore occurrence of seed plants are all important components of embryonic development. The incomplete development of any one of these processes will lead to fertilization failure or reduced seed yield [2]. Through observation of reproductive processes in different crops, it was found that developmental abnormalities at any stage can lead to abortion. During the development of microspores of *Platycladus* arborvitae and kiwifruit, the loss of tapetum and programmed cell death will lead to abnormal meiosis and microspore hollowing, and ultimately abortion [3,4]. During the microsporogenesis of *Koelreuteria elegans*, it was also found that the programmed death of anther wall cells will lead to the failure of anther dehiscence [5]. Scanning electron microscopy observations of the floral development of *Medicago truncatula* [6] and *Astragalus caspicus* [7] revealed that the sequence of floral organ development in each whorl was unidirectional, starting from the back of the flower, with a high degree of overlap. In addition, the presence of abnormal embryo sacs during the development of jujube megaspores was shown to lead to embryo abortion, thereby reducing seed yield [8]; at the same time, observations on the embryonic development of *Magnoliaceae* plants revealed that the loss of synergid cells can also lead to defects in the development of egg cells in the embryo sac [9]. Abnormal pollen development is considered to be one of the main causes of plant abortion. Each stage of microspore development plays an important role in the growth of pollen grains and their successful fertilization [10]. The anthers of Caucasian clover (*Trifolium ambiguum*) have only two pollen sacs, and only 35% of the pollen is viable. In addition, its embryo sac is poorly developed, which ultimately leads to ovule abortion [11].

Embryological characteristics are an important basis for judging the compatibility between plant species and determining the process of plant evolution [12]. There are 11 species in the genus *Gossypium*, most of which have an Allium-type embryo sac development type, but one has a Drusa-type embryo sac with 4 to 11 antipodal cells [13]; after the megasporogenesis of *Trichoderma* and *Dodonaeaeae* (both belonging to the family *Sapindaceae*), the nucleolus in the embryo sac degenerates in the same way as the integument of the ovule [5]. It can be seen that the same species have relatively consistent development patterns, and it can be seen that the male and female gametophytes of closely related species may have similar development patterns [14].

*L. davurica* is a high-quality perennial legume forage, mainly distributed in temperate regions of the world. It is one of the dominant species in northeast, north, and northwest China. Its community mainly occurs in forest and grassland areas. It is a high-quality forage with excellent quality, strong stress resistance, and good ecological effects [15,16,17]. Grass plants have complex inflorescences and solitary flowers. Reproductive development plays an important role in the reproduction of flowering plant populations. Understanding the reproductive development patterns of *L. davurica* and the abnormalities existing in the development of floral organs can provide important reference values for subsequent molecular breeding and improvement of *L. davurica*. However, there are currently few studies on the comprehensive reproductive biology of leguminous plants, including studies on the development of male and female floral organs and observations of the fertilization process. From a molecular perspective, the timing and duration of gene expression during floral organogenesis are fundamental components of the genetic mechanisms that control reproductive development [18,19]. Understanding the temporal expression patterns of organ-specific genes in stamens and pistils can help elucidate the genetic mechanisms controlling the formation of fully functional flowers [20]. Therefore, this study starts from histology and cytology and studies the flower bud differentiation and development of megaspores and microspores and male and female gametophytes of *L. davurica* to provide practical guidance and a theoretical basis for its high-yield cultivation and genetic improvement. Through phylogenetic tree analysis, we provide basic information for the developmental biology and molecular biology of plants in the *Fabaceae* family and especially of the *Lespedeza* genus.

## 2. Results

### 2.1. Microscopic Morphological Observation of Flower Structure

The inflorescences of *L. davurica* are axillary racemes, with 11–19 flowers on each inflorescence (Figure 1A). The flowers are bisexual and bloom in August–September. The flowers consist of a pedicel, receptacle and calyx, corolla, stamens, and pistils. The corolla is a typical butterfly-shaped corolla (Figure 1B), consisting of one flag petal, two wing petals, and two keel petals. Under the electron microscope, pollen can be observed dispersing from the anthers after dehiscence (Figure 1C). Each anther has four pollen sacs, there are fine wart-like protrusions on the surface of the anthers, and the pollen will be released from the pollen sacs when mature. The style is white to transparent, and yellow pollen grains can be clearly seen adhering to the style. Under a scanning electron microscope, a ring of pollen attachment can be observed on the stigma. The style, stigma, and ovary are all densely covered with long, silky hairs (Figure 1D).

### 2.2. Observation of Pollen Germination and Pollen Tube Elongation

The dynamic growth process of *L. davurica* pollen tubes was observed through field artificial pollination combined with fluorescence microscopy. One hour after pollination, it was found that pollen adhered, hydrated, and sprouted into pollen tubes on the stigma. It was observed that several pollen tubes had germinated in the style. The newly germinated pollen tubes were shorter, and a small number of them had callose in the early stages of germination. A small amount of callose can be observed on most stigmas (Figure 2A,B). After 2 h of pollination, the pollen tube had reached 1/3 of the style, displaying irregular growth patterns, including straight and curved growth. Intermittent callose was also observed inside the pollen tubes (Figure 2C). The pollen tubes grew to reach 1/2 of the style 4 h after pollination. The pollen tubes grew in a filament-like curved state, and there was obvious callose at the top of the pollen tubes (Figure 2D). After 6 h of pollination, the pollen tube reached the top of the ovule. Intermittent callose appeared in the pollen tube, and the fluorescence signal was obvious. There was a callose mass in the middle of the pollen tube (Figure 2E). The pollen tubes reached the embryo sac in bundles. There was still callose in the pollen tube 12 h after pollination. The pollen tube entered from the micropyle above the round embryo sac (Figure 2F). Abnormal elongation phenomena were found during the elongation and growth of pollen tubes, such as spiral growth of pollen tubes and discontinuous callose (Figure 2G,H).

### 2.3. Flower Bud Differentiation

The inflorescence type of *L. davurica* is raceme, and its flower bud differentiation is divided into six stages, including vegetative growth stage, flower bud differentiation stage, sepal primordium differentiation stage, petal primordium differentiation stage, stamen primordium differentiation stage, and pistil primordium differentiation stage. Flower buds begin to sprout in early May. At this time, the growth cone is shaped like a smooth sphere, with a raised top and surrounded by flower bracts at both ends. It can be seen that the cells in the growth cone are small and neatly arranged, and the cells are dividing vigorously. This protrusion is the vegetative growth stage of flower bud differentiation (Figure 3A–C). In mid-May, when it has entered the stage of morphological differentiation, the apical meristematic tissue of the growth cone divides vigorously, and the nutrient growth cone begins to expand laterally (Figure 3D). From mid-to-late May, as the apical cells continue to divide, one side of the flattened growth cone will protrude first. As the cells divide in the sepal primordia, the small protrusions formed will move toward the sepal primordium. They will bend inward, eventually forming two symmetrical leaf-like sepals (Figure 3E–G); in late May, as the sepal primordia elongate upward and bend inward, the inner petal primordium also begins to form symmetrical small protrusions, which continue to grow and form a corolla (Figure 3H,I); from the end of May to the beginning of June, the division activity of the reproductive growth cone cells is exceptionally vigorous, and the cells become larger in size; the cells have large nuclei and dense cytoplasm, and the cells are arranged more closely. Multiple small protrusions are formed at the base, and then the stamen primordia differentiates; there is subsequent differentiation of anthers and filaments, and pollen finally differentiates from the four pollen sacs (Figure 3J,K); in early June, while the stamen primordium differentiates, the pistil primordium also begins to differentiate, so the development of stamen primordium and pistil primordium can overlap in time. The pistil primordium gradually transitions from small protrusions to vertical growth to form the pistil, and the hollow ovary at the bottom also differentiates into an anatropous ovule (Figure 3L–N).

### 2.4. Microsporogenesis and Male Gametophyte Formation

The development of the anther wall goes through a series of complex processes from anther primordium to mature anther. *L. davurica* has six stamens, and each anther contains four microsporangia. When the anther is in the anther primordium stage, most of cells in tissues have a strong ability to divide (Figure 4A). The cells in the four corners divide vigorously, producing the initial archesporial cell (Figure 4B,C). After archesporial cell periclinal division, the primary parietal layer and primary sporogenous cells are produced. The primary sporogenous cells divide multiple times to form secondary sporogenous cells (Figure 4D). The primary parietal layer undergoes several circumferential divisions to form a complete anther wall structure, which consists of epidermis, endothecium, middle layers, and tapetum from outside to inside (Figure 4E). It can be seen that the anther wall development type of *L. davurica* is the dicotyledon type. There is only one layer of cells on the endothecium of the anther (Figure 4F,G). When the pollen matures, only the endothecium of the anther wall remains; the middle layers are squeezed by other anther wall tissues during meiosis, and their shape becomes flat, and then they are absorbed or disintegrated (Figure 4H). When tapetum cells are first formed, their cytoplasm is thick, and their cells are flat. During the microspore mother cell meiosis, the tapetum cell nucleus undergoes mitosis and becomes binucleate. Later, during microspore meiosis division, the cells are elongated and rectangular, with large nuclei and dense cytoplasm. In the free microspore stage, the shape of the tapetum cells changes from a quadrilateral to an irregular triangle, with blurred cell edges, and the nucleus degenerates and disintegrates. In the anther walls of mature pollen, only a few tapetal cells remain, or they disappear completely (Figure 4I–K).

The microspore mother cells in *L. davurica* are differentiated from primary sporogenous cells. The microspore mother cells are larger in size and have obvious nucleoli and no obvious vacuoles (Figure 5A). In the prophase of meiosis, the chromosomes in the cytoplasm of the microspore mother cell are spirally coiled and gradually become filamentous. After the first meiotic division, two nuclei are produced without forming a cell wall (Figure 5B–E), and during the second meiosis, two nuclei divide simultaneously and the cell wall is produced, forming a tetrad (Figure 5F–I). The arrangements of the tetrads include left–right symmetry and are tetrahedral (Figure 5H,I). The four newly formed microspores are located within the callose wall. When the callose wall dissolves, the microspores are released to form free microspores (Figure 5J). It can be seen that the meiosis mode of microspores of *L. davurica* is cytoplasmic simultaneity. After aniline blue staining of microspore mother cells, it was found that there was no fluorescence between the four nuclei formed after the second meiosis, indicating no callose wall formation (Figure 5K), and at a later stage when the four nuclei start to separate, a blue fluorescence between the nuclei can be clearly seen (Figure 5L), indicating that callose walls have formed between the nuclei at this time. Mature pollen has no callose.

The free microspores of *L. davurica* form mononuclear pollen, which is free in pollen sacs. Its external shape is horseshoe-shaped, with shriveled cells, thin cell walls, and thick cytoplasm, and the nucleus is located in the center of the cell (Figure 5M,N); as the pollen absorbs nutrients from the tapetum, a large vacuole appears in the center of the pollen, and the cell nucleus is squeezed to the edge of the pollen wall. This stage is the mid–late uninucleate stage (Figure 5O). After the first mitotic division, mononuclear pollen forms two cells of different sizes, namely two-cell pollen (Figure 5P); the larger one is the vegetative nucleus, and the smaller one is the reproductive nucleus; pollen is released from the dehisced anthers after maturity (Figure 5Q,R).

### 2.5. Megasporogenesis and Formation of Female Gametophyte

The cells under the epidermis in the ovary wall of *L. davurica* undergo periclinal division to produce the ovule primordium, with the nucellus at the top (Figure 6A), an annular protrusion differentiated above the nucellus epidermis, which is divided into an inner integument and outer integument. The proliferation and differentiation rate of inner integument cells is higher than those of the outer integument. Eventually, the inner integument wraps the entire nucellus (Figure 6B,C), leaving a small hole at the top, which is the micropyle, and the ovule growth pattern is anatropous. The megaspore mother cell is formed from the sporogenous cells below the nucellar epidermis. It has thick cytoplasm and a prominent nucleus. The megaspore mother cell is surrounded by the surrounding callose wall (Figure 6D), and undergoes two meiotic divisions, forming a tetrad (Figure 6E). The four megaspores in the tetrad are arranged in a linear manner, and the three megaspores near the micropyle are surrounded by callose walls (Figure 6E) and degenerated and disappeared due to a lack of nutrients required for growth, and finally only the megaspore at the chalazal end is left (Figure 6F), which forms a single-nucleated embryo sac, so the embryo sac development type of *L. davurica* is monosporous.

The megaspore at the chalaza develops into a functional megaspore, a single-nucleated embryo sac, and then undergoes the first mitotic division to form two nuclei (Figure 6G). The two nuclei are initially arranged laterally (Figure 6H), and then move toward the chalaza and micropylar end of the embryo sac, respectively, forming a dinuclear embryo sac (Figure 6I). Subsequently, after several mitotic divisions, a four-nucleated embryo sac (Figure 6J) and an eight-nucleated embryo sac (Figure 6K–M) are formed. At this time, there are four nuclei each at the micropyle and chalazal end. Then, one nucleus each at the micropyle and chalazal end move toward the middle of the embryo sac, forming a central cell with two nuclei (Figure 6N). The three nuclei at the micropyle end form the egg apparatus, which contains one egg cell and two synergid cells (Figure 6O), and the three nuclei at the chalazal end form three antipodal cells (Figure 6P). At this point, the embryo sac of *L. davurica* has formed a mature state of seven cells and eight nuclei. In the mature embryo sac, the nucleus of the egg cell is biased toward the side wall of the embryo sac (Figure 6Q); the cytoplasm is closer to the chalazal end, and there is an obvious filiform apparatus at the bottom of the synergid (Figure 6R). The number of nuclei of the antipodal cells at the chalazal end gradually increases from three to several during the development process (Figure 6S). It can be seen that the development type of the embryo sac of *L. davurica* is a monosporous polygonum type.

### 2.6. Double Fertilization, Embryo and Endosperm Development

*L. davurica* pollen sprouts on the stigma, and the pollen tube is guided by the style to the top of the ovule. After entering the embryo sac, the pollen tube ruptures, and the released sperm cells fuse with the egg cell and polar nucleus, respectively. The style is hollow with a style canal in the middle, there is a layer of inner epidermal cells surrounding the stylar canal (Figure 7A,B).

It can be seen from the section that fertilization of the egg cell occurs earlier than the fertilization of the polar nucleus, but because the zygote has a dormant period, the formation of the primary endosperm nucleus occurs earlier than that of the fertilized egg. The content released by the pollen tube contains two sperm. The sperm is large in size and has an obvious nucleus. The sperm and the egg cell meet at the chalazal end of the embryo sac (Figure 7C), and the plasma membranes of the two cells fuse (Figure 7D), the distance between the nuclei of the sperm cell and the egg cell shortens, and then the nuclear envelope and nucleoplasm of the two cells fuse (Figure 7E,F); eventually, they fuse into one cell nucleus to form a fertilized egg, namely the zygote. The fertilization process of the other sperm from the pollen tube with the polar nucleus is similar to the fertilization process with the egg cell. The sperm in the embryo sac (Figure 7G) enters the polar nucleus and is close to its nucleolus. After the fusion is completed, the primary endosperm nucleus is formed (Figure 7H), and there is no cell wall between the nuclei of the endosperm primitive cells. This stage is called the free endosperm nucleus stage (Figure 7I,J). As their number increases, cell walls form between free nuclei, thus forming endosperm cells (Figure 7K). Free endosperm cells form a circle within the embryo sac, forming a peripheral free cell layer (Figure 7L,M); as the embryo sac becomes larger, free endosperm cells increase. When the zygote develops into a torpedo-shaped embryo, the endosperm begins to degenerate until it disappears. When the embryo is fully mature, there are no endosperm cells in the embryo (Figure 7N).

The zygote of *L. davurica* begins to divide after a period of dormancy. The zygote undergoes the first transverse division to form one basal cell and one apical cell (Figure 8A). After multiple consecutive transverse divisions, the basal cell forms a suspensor consisting of 5 to 8 cells (Figure 8B,C). The apical cell undergoes two longitudinal divisions to form the four cells, which is the tetrad stage (Figure 8D),then each cell undergoes a lateral division to form eight cells, which is the octagonal stage (Figure 8E). The eight-part somatic cells undergo pericircular division to form a spherical tissue, with a total of 32 cells. This is called the spherical embryo stage (Figure 8F–H), and the growth and differentiation of the proembryonic stage are completed. The cells on both sides of the spherical embryo body divide vigorously, forming two protrusions, called cotyledon primordia. As the protrusions become larger, they take on a heart shape, which is called the heart-shaped embryo stage (Figure 8I,G). With the rapid growth of the cotyledon primordium and the inconsistency of cell division, the cotyledon part of the embryo becomes curved due to different cell division speeds (Figure 8K,L), while the cells in the hypocotyl area divide vigorously and grow in a straight line, called the torpedo-shaped embryo stage (Figure 8M,N); later, the embryo body matures and the embryonic suspensor gradually disappears, and protruding germ tissue can be seen in the mature embryo (Figure 8O).

### 2.7. Structure and Development Process of Mature Seeds

Each ovary of *L. davurica* has only one ovule. Each flower forms one seed (Figure 9A). The seeds are often wrapped in the persistent calyx. Mature seeds are obovate. The epidermis of the seeds is reticulated and convex on both sides. The seed coat is green in the early stage and turns yellow-brown when mature. There are seed holes and hilum on the surface of the seed coat (Figure 9B,C). Mature seeds are composed of seed coat, hypocotyl, embryo, and cotyledons. The cotyledons occupy most of the space of the seed. The hypocotyl and embryo are located on the sides of the seed. The dormancy period of seeds is longer, and generally the seed hardening rate is higher that year. During the germination process, the seeds first absorb water and swell. After the seed coat ruptures, the radicle first breaks through the seed coat (Figure 9D,E). Root hairs have grown on the radicle, and finally, the radicle will develop into the root system. The hypocotyl on the radicle serves to connect the embryo and the radicle. The elongation of the hypocotyl pushes the embryo out of the seed coat, the embryo then develops into stems and leaves (Figure 9F,G), and the cotyledons serving as nutrient organs gradually disappear.

### 2.8. Phylogenetic Tree Analysis

To investigate the phylogenetic relationship between *L. davurica* and other plants within *Fabaceae* and *Poaceae*, we developed a comprehensive system using genomic data from species available in the National Center for Biotechnology Information (NCBI) database. Our study included *L. daurica*, *Arabidopsis thaliana*, *Oryza sativa*, *Zea mays*, *Medicago sativa*, and species of the *Lespedeza* and *Medicago* genera.

It is worth noting that the phylogenetic tree is divided into two branches, namely clade I and clade II. Clade I includes *Arabidopsis*, rice, and corn, and clade II can be divided into three subclades, including *Medicago*, soybean, and *Lespedeza* (Figure 10). The genus *Medicago* includes *Medicago archiducis*, *Medicago sativa*, *Medicago polymorpha*, *Medicago lupulina*, *medicago truncatula*; the genus *Lespedeza* includes *Lespedeza davurica*, *Lespedeza buergeri*, *Lespedeza chinensis*, *Lespedeza davidii*, *Lespedeza floribunda*, *Lespedeza fordii*, *Lespedeza Pilosa*, *Lespedeza potaninii*, *Lespedeza virgata*, *Lespedeza inschanica*.

## 3. Discussion

Embryology involves the study of anther, ovule, and seed development; more than 50 characters are provided for the analysis of family, genus, and intraspecific and interspecific phylogeny [21]. In this study, paraffin sectioning and staining techniques were employed to systematically compare the developmental timing of male and female flower organs in *L. davurica*. It takes 4 months from the flower bud differentiation to finally form mature seeds. During this period, *L. davurica* megaspores and microspores undergo meiosis, resulting in the formation of male and female gametes, respectively. Finally, the mature pollen germinates, and the pollen tube enters the embryo sac through the style tract. After double fertilization is completed, the zygote undergoes multiple mitotic divisions to form a mature embryo.

### 3.1. Flower Bud Differentiation

The differentiation of flower buds is the prerequisite for flower formation, and the transformation process from leaf buds to flower buds is an indispensable link in the formation of flower organs [22]. Nevertheless, the duration of floral bud differentiation exhibits significant variation across plant species [23,24]. Differences in flower bud differentiation stages of leguminous plants have been reported, but there is no clear standard to divide flower bud differentiation stages. The arrangement and development of floral organs is unidirectional. The outer ring develops first and then the inner ring matures [25,26]. Research shows that understanding the characteristics and stages of plant flower bud differentiation and ensuring the quality and quantity of plant flower buds are of great significance to production [27].

This study found the flower bud differentiation stage of *L. davurica* lasted about 40 days and was divided into six stages. The differentiation was completed in mid-June. The growth cone differentiated first to form the stamen primordium, and the pistil primordium formed later; this is the same sequence of flower bud differentiation found in other species such as camellia [28], magnolia [29], tobacco [30]. Among them, the differentiation time of different flower buds exhibits little difference in different individual plants, and different differentiation stages overlap in the same inflorescence and different plants. Among leguminous dicotyledonous plants, the flowering time of most plants is from August to September. It can be seen that the flower bud differentiation time of *L. davurica* is longer and the differentiation speed is slower. It is speculated that the flower bud differentiation time is longer to ensure that the final number of florets meets the pollination demand.

### 3.2. Development of Male Gametophyte

Studies on anther wall cells of different species of angiosperms have divided the development patterns of anther walls into four types: basic, dicotyledonous, monocotyledonous, and simplified [31]. According to the classification of anther wall formation types by Dias [32], the anther wall development type of *L. davurica* belongs to the dicotyledonous type. The development of microspores is closely related to the tapetum structure. Tapetum cells can produce callose enzyme to destroy the callose wall in the tetrad stage of microspores and release microspores [33]; the tapetum provides energy for microspores during development and then degrades into irregular cell clusters [34]; abnormal tapetum structure is not conducive to anther maturation and leads to pollen abortion [35,36]. The cell shape and decomposition rate of the tapetum will also affect the growth of microspores, which in turn affects the viability of pollen [37]. Therefore, the development of microspores is closely related to the normal decomposition of the tapetum. We found the tapetum of *L. davurica* remained unchanged during the development of the anther wall and belonged to the glandular tapetum, and the tapetum did not undergo abnormal changes at various stages of microspore differentiation and development. Under normal circumstances, the meiosis process of each microspore mother cell should be highly synchronous [28]. We have observed that the division process of microspores in the same flower or the same anther chamber showed asynchronous characteristics. This is consistent with the observations on microspore meiosis in *Lagerstroemia speciosa* [38]. It is speculated that the mechanism of asynchronous differentiation of plants may be to extend the pollination time of plants and improve their adaptation to the environment, and provide a guarantee for the successful reproduction of the species, which may have a certain positive significance for its evolution [38].

### 3.3. Elongation of Pollen Tubes

The molecular recognition of pollen grains and stigma surfaces promotes pollen adhesion and germination. The secretion secreted by the wet stigma is crucial for the adhesion and germination of pollen grains. The subsequent growth and guidance of the pollen tube is influenced by the sporophytic and gametophytic systems [39]. Previous researchers observed the growth of pollen tubes in the stigma of *Camellia oleifera* [28] after self-pollination and cross-pollination, and found that the pollen tubes after self-pollination grew abnormally in the style, showing self-incompatibility. Fluorescence microscopic observation revealed that there were many pollen grains attached to the stigmas of *L. davurica*, but there are very few pollen tubes in the style.

The limited affinity between stigma and pollen may prevent most pollen from hydrating on the stigma for germination. Furthermore, the presence of callose causes pollen tube growth arrest as pollen tubes elongate in the style, indicating the gametophyte incompatibility of *L. davurica*. Callose is a β-1,3-glucan that covers the cell walls of pollen tubes. It plays a crucial role in various stages of plant growth and development. It can reduce the cell turgor pressure caused by the elongation of the top of the pollen tube, and it can also increase the ductility of the pollen tube tip to maintain the growth of polar pollen tubes [40]. In addition, callose is deposited at the site of pathogen invasion, between the plasma membrane and the cell wall in plasmodesmata and other plant tissues, to slow down the invasion and spread of pathogens [41]. In addition, its excessive accumulation during pollen tube elongation can hinder pollen tube growth. However, there remains a lack of a definitive basis regarding the sequence of pollen tube stasis and callose plug production. Recent studies on the molecular mechanism of fertilization in *Arabidopsis* have revealed that upon the pollen tube’s entry into the ovule and successful identification with the egg cell, the FER (FERONIA), ANJ (ANJEA), and HERK1 (HERCULES RECEPTOR KINASE 1) receptor complex proteins located at the micropyle end cooperate to regulate pollen tube growth, thereby preventing polyspermy and ensuring reproductive success [42]. In this experiment, we postulated that the initial arrest in pollen tube growth might result from the entry of existing pollen tubes into the ovule. Consequently, the reproductive system suppresses the growth of additional pollen tubes to prevent polyspermy and ensure successful fertilization. Research has revealed that FER can interact with a small molecule peptide known as RALF (Rapid Alkalinization Factor) and modulate the cellular pH environment [43]. This interaction ultimately leads to the stasis and rupture of pollen tube growth. Additionally, chaperone proteins called RLKs (Receptor-Like Kinases) play a role in regulating pollen tube rupture. Notably, loss-of-function *Arabidopsis* mutants of *RLKs* exhibit an overgrowth phenotype in pollen tubes, which subsequently fail to release sperm cells upon entering the embryo sac [44,45]. The observed abnormal phenomena, such as the bending and winding of pollen tubes within the style in our study, may arise from alterations in the growth environment of the pollen tube, resulting in the expression of specific proteins. Further investigation is necessary to elucidate the precise reasons behind the changes in pollen tube growth shape.

### 3.4. The Formation of the Female Gametophyte

The female gametophyte plays a critical role in essentially every step of the reproductive process. During pollen tube growth, the female gametophyte participates in directing the pollen tube to the ovule, and induces embryonic development after fertilization [46,47]. Female gametophyte development consists of two stages: megasporogenesis and female gametophyte development. Spores develop into gametophytes through cell proliferation and differentiation [30]. During megasporogenesis, the diploid megaspore mother cell undergoes meiosis to produce four haploid nuclei. In our study, *L. davurica* ovules exhibited anatropous development, with the megaspore mother cell producing four megaspores following two meiotic divisions. Ultimately, only the chalazal megaspore developed into a single-nucleated embryo sac. Earlier cytological studies revealed that the chalazal-end megaspore mother cell suppresses the development of the remaining three megaspore mother cells after their differentiation. However, the molecular mechanisms responsible for the degradation of these other three mother cells remain poorly understood. In this experiment, the female gametophyte underwent three meiotic divisions to form a mature state of seven cells and eight nuclei.

The gametophyte primarily consists of egg cells, antipodal cells, and synergid cells. Its embryo sac development follows a monosporous polygonum type. Currently, the polygonum-type development pattern is observed in female gametophytes of over 15 species, including several economically significant crops such as *Brassicaceae* (e.g., *Arabidopsis*, *Capsella*, *brassicas*), *Gramineae* (e.g., corn, rice, wheat), and *Fabaceae* (e.g., beans, soybeans) [48,49,50].

Willemse and van Went [49] observed that the intracellular organelles and functional activities of plant megaspores and microspores are similar before undergoing functional differentiation. However, the developmental patterns of megasporogenesis and microsporogenesis are typically unsynchronized. In this study, microspores developed earlier than megaspores, which is consistent with the order of macrospore development in tobacco [30]. In *Paspalum rufum* [51], male and female reproductive development is synchronous in the diploid stage, whereas the occurrence of megaspores lags behind that of microspores in the tetraploid stage. However, the timing difference in the development of large and small spores does not impact pollination, as sexual maturation is synchronized at the flower level. In this experiment, although the differentiation timing of pistils and stamens in *L. davurica* was not synchronized, both male and female organs matured before the florets opened. It is speculated that this asynchronous development of pistil and stamen in plants can effectively prolong the pollination cycle [52]. Therefore, no abnormal development was observed during the development of male and female gametophytes in *L. davurica*.

### 3.5. Double Fertilization and Early Seed Development

Pollen germinates on the stigma, traverses the style, and reaches the ovule, eventually entering the embryo sac through the micropyle. Subsequently, the pollen tube ruptures, releasing sperm cells. At the angiosperms, two fertilization events occur [53,54]: one sperm combines with the egg cell, initiating embryogenesis by forming a fertilized egg; the other sperm fuses with the central cells in the embryo sac, giving rise to the endosperm. The endosperm persists throughout seed development, providing essential nutrients for the developing embryo and seedlings [55]. The fertilization of the egg cells of *L. davurica* occurred earlier than the fertilization of the polar nuclei; after fertilization, the egg cell formed a zygote, which then entered a dormant state. Subsequently, a sperm cell fused with the central cell to generate an endosperm cell. The endosperm cells underwent mitosis, while the zygote initiated division, eventually forming a spherical, heart-shaped, and torpedo-shaped embryo. During later zygote development stages, endosperm cells degraded and vanished. Mature embryos exhibited no discernible free endosperm tissue. This study reveals that the endosperm cells in *L. davurica* exhibit a developmental pattern consistent with other leguminous plants. These endosperm cells primarily serve as a nutrient source for embryo differentiation and growth. Research shows that approximately 15,000 distinct genes are active during embryo development across various plant species, including soybean and cotton [56]. Most of these genes are expressed in specific differentiation stages, regions, and organs of the embryo, providing a useful entry point into uncovering the molecular mechanisms that regulate cell differentiation and region-specific differences. In addition, studies in legumes and many other plants have found that embryos produce auxin, with the highest levels of auxin occurring during the globular stage of embryonic differentiation; asymmetric auxin distribution is beneficial to the differentiation and formation of heart-shaped embryos [57]. In this experiment, the transition of the embryo from a spherical to a heart-shaped structure is likely driven by asymmetric auxin distribution. Subsequently, as the endosperm cells fully vanish, the embryo attains maturity and becomes a seed. During the early germination phase, the mature seed absorbs water, leading to swelling, followed by the seed coat’s expansion; upon rupture, the hypocotyl penetrates the seed coat, facilitating the emergence of the embryo, which eventually grows into stems and leaves.

### 3.6. Phylogenetic Implications

A phylogenetic tree, commonly used in biology, visually represents the evolutionary relationships among various species, which can illustrate both the genetic relationships and the evolutionary history among distinct species. Phylogenetic trees, which serve as representations of species relationships, are constructed using diverse data sources such as molecular genetics, morphology, and ecology. These phylogenetic trees allow us to infer evolutionary relationships by comparing DNA sequences, morphological features, and physiological traits across different species.

There are great individual differences in growth and development between *Fabaceae*, *Poaceae,* and *Arabidopsis* [6,11,58]. Our phylogenetic analysis reveals that *L. davurica* may have great differences in reproductive development from *Arabidopsis thaliana*, rice, and corn. In addition, the phylogenetic tree shows that *L. davurica* belongs to the same branch as alfalfa, soybean, and *Lespedeza*, indicating that their genetic relationship is close. The processes of flower bud differentiation, tapetum degeneration, and megaspore development of *L. davurica* are consistent with the development patterns of leguminous plants such as alfalfa and clover [6,11].

## 4. Materials and Methods

### 4.1. Growth and Sample Collection of L. davurica

The *L. davurica* cv ‘JinNong 1’ is cultivated in experimental fields in Taigu City, Shanxi Province, China (112°34′ E, 37°21′ N, altitude 800 m). The experimental station’s *L. davurica* flowering period is from June to August. In May to July of 2019 and 2020, samples were taken based on tissue size. During the early growth stage, newly formed flower buds were collected every 2 days, and during the later growth stage, samples were taken every 4 days until the end of flower bud differentiation. Each time, 30 flower buds were collected and preserved in a fixative solution of 50% FAA (50% ethanol, 10% formaldehyde, 5% acetic acid). Sampling for the development of male and female gametophytes occurred from early June to late August. Developmental stages were distinguished based on tissue length. After removing sepals, samples were fixed in 50% FAA for at least 24 h. Sampling for the development of fertilization, embryos, endosperms, and seeds took place from mid-August to the end of September. After the stigma protruded, different open flowers and pods were selected. After removing the outer skin, samples were fixed in 70% FAA.

### 4.2. Paraffin Section Preparation and Observation

The preparation for paraffin section samples followed the method described by Williams et al. [59] and González-Melendi et al. [10]. After the collected samples were fixed in FAA for at least 24 h, they were dehydrated in 30% and 50% alcohol for 1 h each. Subsequently, the samples were dyed with hematoxylin, with young samples stained for 48 h and other samples stained for 72 h. After the whole staining, the samples were rinsed with running water for 24 h. Following dehydration in a gradient of 30%, 50%, 70%, 85%, 95%, and 100% alcohol, the samples underwent transparent treatment using a mixture of ethanol and xylene and pure xylene. Finally, the samples were embedded in paraffin, cut into thin slices of 8 μm (Leica RM2235); the slides with tissue samples were dried and then made transparent using xylene; after staining with safranin-fast green, the slides were mounted, and the sections were observed and photographed under a microscope (OLYMPUS BX51). The hematoxylin staining resulted in the cell nucleus and chromosomes appearing in dark blue, while safranin stained the cell nucleus and lignified cell walls red, and fast green stained the cytoplasm green.

### 4.3. Microscopic Observation of Morphological Development of Flower Buds and Florets

Small flower samples were collected at 8, 24, 36, 48, and 72 h after pollination. The pistils were fixed in Carnoy’s fixative (alcohol/acetic acid = 3:1) for 24 h and then transferred to 70% alcohol. The samples were soaked in a 58 °C 8 mol/L NaOH solution for 3–5 h, followed by staining in a 5% aniline blue solution prepared with 1% K_2_HPO_4_ for 4–6 h. After slide preparation, observation and analysis were conducted under a fluorescence microscope, specifically the Leica DMC6200 model, to study ovule development and pectin deposition.

### 4.4. SEM Observation

We collected flower buds and flowers during the flowering period, and removed petals and sepals for observation under a dissecting microscope and scanning electron microscope (SEM). The preparation of scanning electron microscope samples followed the methods of previous researchers [28]. Initially, the samples were fixed in 2.5% glutaraldehyde–phosphate buffer for 24 h. After washing three times with phosphate buffer (pH = 7.2), the samples were dehydrated in a gradient of 30%, 50%, 70%, 80%, 90%, and 100% ethanol. After two washes with tertiary butanol, the samples were cooled in a refrigerator at 4 °C. Subsequently, the samples were placed in a vacuum coater bell jar, subjected to 1 h of vacuuming, and then affixed to small iron pieces for gold coating. Finally, we observed them under the Hitachi JSM-6490V field emission scanning electron microscope.

### 4.5. Phylogenetic Tree

We searched the NCBI (National Center of Biotechnology Information) online website (https://www.ncbi.nlm.nih.gov/Taxonomy/CommonTree/wwwcmt.cgi, accessed on 10 April 2024) using the plant Latin names of *Arabidopsis thaliana*, rice, corn, *Lespedeza,* and *Medicago*, saved the searched species information in the phylip tree format, and used the online tool iTOL to beautify it.

## 5. Conclusions

In summary, our study is the first systematic study on the embryological development process of the *Lespedeza* genus. We found that pistils develop later than stamens during flower bud differentiation in *Lespedeza*, but the sexual maturation of male and female flower organs is synchronous throughout the development of florets. Additionally, we observed that pollen tubes exhibit bending and growth cessation as they traverse the style. The differentiation changes in male and female floral organs at different developmental stages in this study provide a morphological basis for reproductive differences among other leguminous plants. In addition, the developmental abnormalities displayed by pollen tubes during growth can serve as an entry point for other leguminous plants to improve seed setting rates through molecular mechanisms.

## Figures and Tables

**Figure 1 plants-13-01661-f001:**
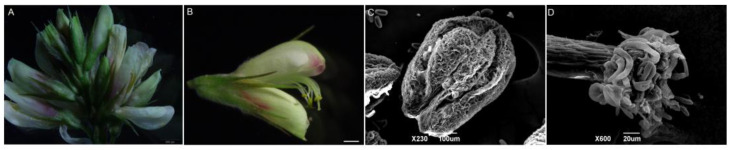
Organ morphological anatomy of *L. davurica*. (**A**) Complete inflorescence, including florets at different stages of development. Scale bar = 500 μm. (**B**) Structural composition of complete florets. Scale bar = 100 μm. (**C**) Scanning electron microscope image of the anther of *L. davurica*, showing fine wart-like protrusions between two pollen sacs. Scale bar = 100 μm. (**D**) Scanning electron microscope image of the stigma, with finger-like hairs at the top where pollen adheres. Scale bar = 20 μm.

**Figure 2 plants-13-01661-f002:**
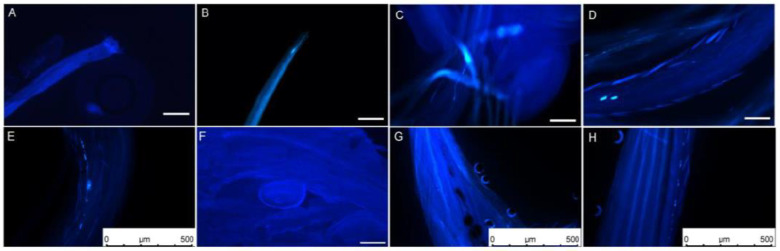
Elongation process of *L. davurica* pollen on the stigma after artificial pollination. (**A**,**B**) Fluorescence micrographs of the stigma’s top one hour after pollination. Pollen aggregates at the top of the stigma but has not germinated. Scale bar = 100 μm. (**C**) Fluorescence micrograph of pollen tube elongation within the stigma two hours after pollination. Obvious callose fluorescence can be seen at the front end of the pollen tube. Scale bar = 100 μm. (**D**) Fluorescence micrograph of pollen tube elongation within the stigma 4 h after pollination. Scale bar = 100 μm. (**E**) Fluorescence micrograph of the pollen tube elongating in the style 6 h after pollination. There is an obvious callose mass in the style tract. Scale bar = 500 μm. (**F**) Fluorescence micrograph of pollen tube reaching above the ovule 12 h after pollination. Scale bar = 100 μm. (**G**,**H**) Anomalies observed during the pollen tube elongation process within the style, such as spiral elongation and callose plugs. Scale bar = 500 μm.

**Figure 3 plants-13-01661-f003:**
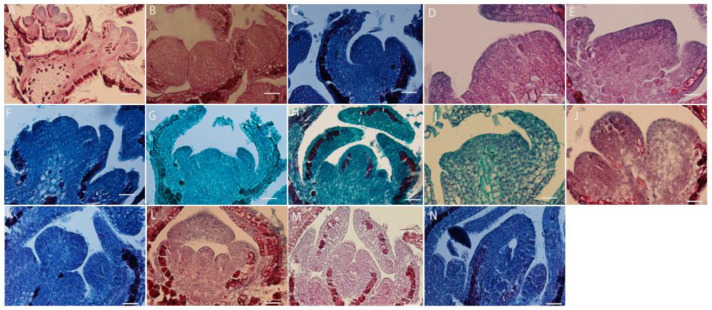
*L. davurica* flower buds differentiate to form floral organ tissues. (**A**–**C**) Flower bud differentiation and vegetative growth stages. Sepals differentiate at both ends of the growth cone. Scale bar = 100 μm. (**D**) Flower bud differentiation stage. Flower buds differentiate laterally. Scale bar = 100 μm. (**E**–**G**) Morphological differentiation stage of the bud, with the formation of sepals. Scale bar = 100 μm. (**H**,**I**) Formation of petal primordia. Scale bar = 100 μm. (**J**,**K**) Formation of stamen primordia. Scale bar = 100 μm. (**L**–**N**) Formation of carpel primordia, with ovules differentiating at the base. Scale bar = 100 μm.

**Figure 4 plants-13-01661-f004:**
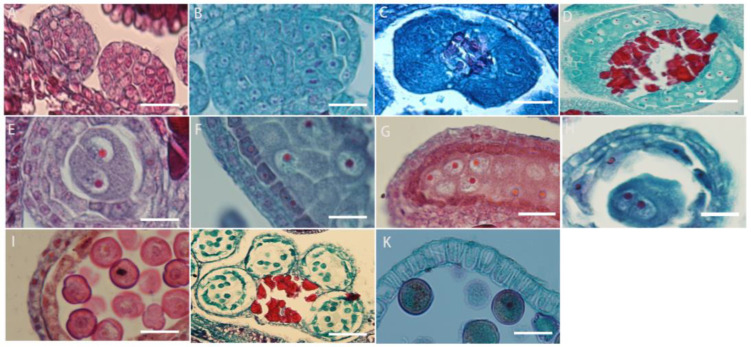
Microscopic observation of anther wall formation of *L. davurica*. (**A**) Anther primordium stage. Scale bar = 100 μm. (**B**,**C**) Prosporial cells. Scale bar = 100 μm. Pericycle division produces primary peripheral cells and primary sporogenous cells. Scale bar = 100 μm. (**D**) Secondary sporogenous cells. Scale bar = 100 μm. (**E**) Complete anther wall tissue, including the inner wall of the anther chamber, the middle layer of cells, and the tapetum. Scale bar = 100 μm. (**F**,**G**) Anther lining cells. Scale bar = 100 μm. (**H**) The middle cells disintegrate and disappear when pollen matures. Scale bar = 100 μm. (**I**–**K**) The tapetum begins to degenerate and disintegrate at the end of microspore development. Scale bar = 100 μm.

**Figure 5 plants-13-01661-f005:**
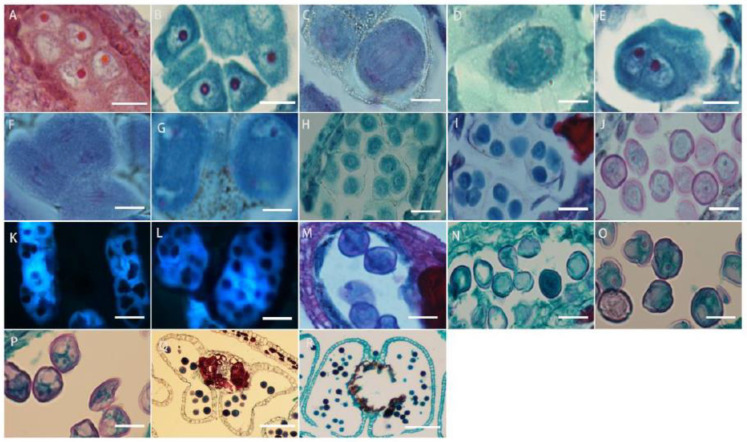
Microscopic observation of microspore development of *L. davurica*. (**A**) Microspore mother cell. Scale bar = 100 μm. (**B**–**E**) The first meiotic division of the microspore mother cell without cell wall formation. Scale bar = 100 μm. (**F**–**I**) Second meiotic division of the microspore mother cell to form a tetrad. Scale bar = 100 μm. (**J**) Free microspores. Scale bar = 100 μm. (**K**) No cell wall is formed in the tetrad. Scale bar = 100 μm. (**L**) Fluorescence microscopy reveals obvious callose wall formation in microspores. Scale bar = 100 μm. (**M**,**N**) Mononucleate pollen is free in pollen sacs. Scale bar = 100 μm. (**O**) Nucleus edge stage in pollen. Scale bar = 100 μm. (**P**) Two-cell pollen stage. Scale bar = 100 μm. (**Q**,**R**) Mature pollen is shed. Scale bar = 100 μm.

**Figure 6 plants-13-01661-f006:**
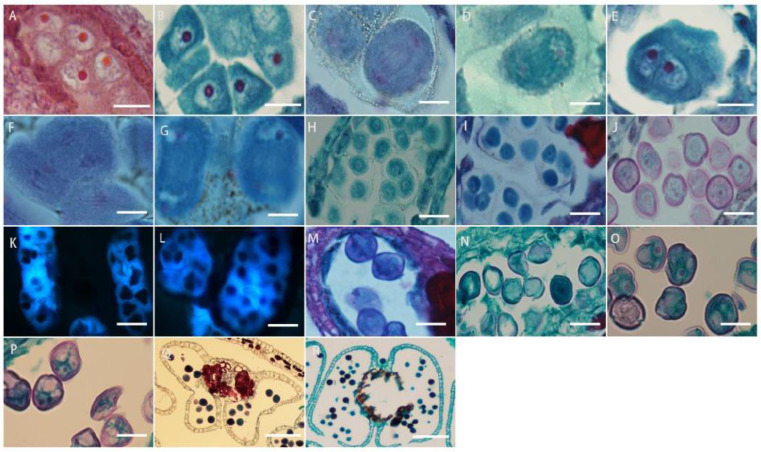
Microscopic observation of megasporogenesis and female gametophyte formation in *L. davurica*. (**A**) Nucellus formation. Scale bar = 100 μm. (**B**,**C**) Differentiation of inner and outer integuments. Scale bar = 100 μm. (**D**) Megaspore mother cell formation. Scale bar = 100 μm. (**E**) Tetrads are formed after 2 meiotic divisions. Scale bar = 100 μm. (**F**) Megaspores in tetrad arranged linearly. Scale bar = 100 μm. (**G**) Megaspore at the chalazal end. Scale bar = 100 μm. (**H**) The first mitotic division of the mononuclear blastocyst forms 2 cells. Scale bar = 100 μm. (**I**) Nuclei are arranged laterally. Scale bar = 100 μm. (**J**) Four-nucleated embryo sac. Scale bar = 100 μm. (**K**–**M**) Eight-nucleated embryo sac. Scale bar = 100 μm. (**N**) Three antipodal cells. Scale bar = 100 μm. (**O**) Seven-cell, eight-nucleus mature embryo sac. Scale bar = 100 μm. (**P**) Filamentous organ at the base of synergid cells. Scale bar = 100 μm. (**Q**) Increased number of antipodal cells. Scale bar = 100 μm.

**Figure 7 plants-13-01661-f007:**
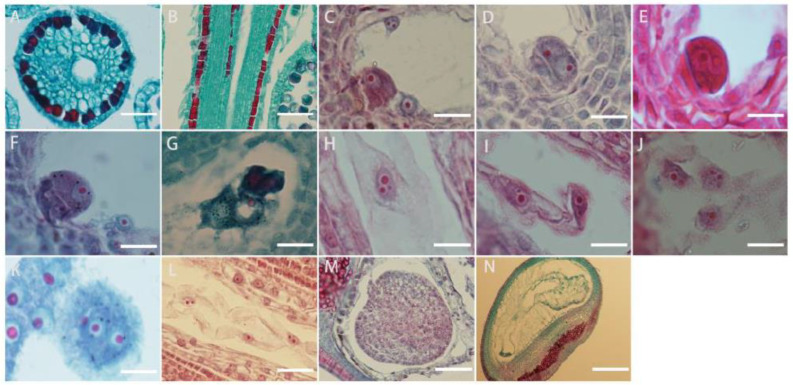
Microscopic observation of double fertilization process in *L. davurica*. (**A**,**B**) Style cross and side section. Scale bar = 100 μm. (**C**) The sperm and egg cells are at the chalazal end. Scale bar = 100 μm. (**D**) The fusion of the plasma membranes of sperm and egg cells occurs. Scale bar = 100 μm. (**E**,**F**) Formation of the zygote. Scale bar = 100 μm. (**G**–**I**) Formation of the primary endosperm nucleus. Scale bar = 100 μm. (**J**) Free endosperm nuclear stage. Scale bar = 100 μm. (**K**,**L**) Formation of endosperm cells. Scale bar = 100 μm. (**M**) Formation of the free cell layer. Scale bar = 100 μm. (**N**) The endosperm undergoes degeneration and disappears. Scale bar = 100 μm.

**Figure 8 plants-13-01661-f008:**
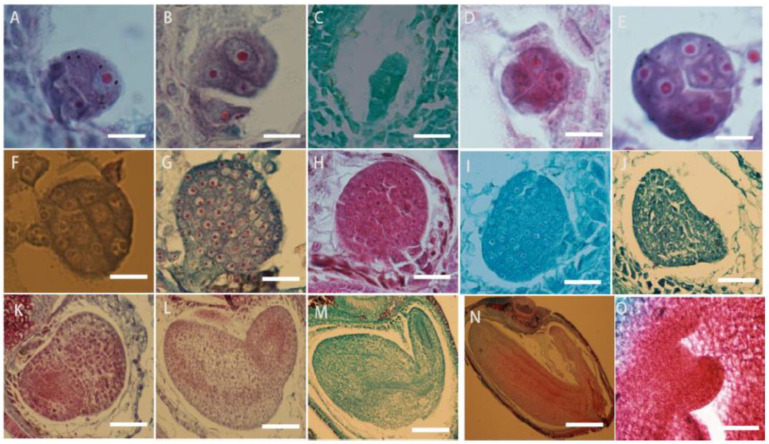
Microscopic observation of the zygotic developmental process in *L. davurica*. (**A**) Formation of one basal cell and one apical cell. Scale bar = 100 μm. (**B**,**C**) Formation of embryonic suspensor. Scale bar = 100 μm. (**D**) Retrad stage. Scale bar = 100 μm. (**E**) Formation of an 8-split period. Scale bar = 100 μm. (**F**–**H**) Formation of the globular embryo stage. Scale bar = 100 μm. (**I**,**J**) Formation of the heart-shaped embryo stage. Scale bar = 100 μm. (**K**,**L**) Uneven differentiation and curvature of cotyledons occur. Scale bar = 100 μm. (**M**,**N**) Formation of the torpedo embryo stage. Scale bar = 100 μm. (**O**) Formation of the germ in the mature embryo. Scale bar = 100 μm.

**Figure 9 plants-13-01661-f009:**
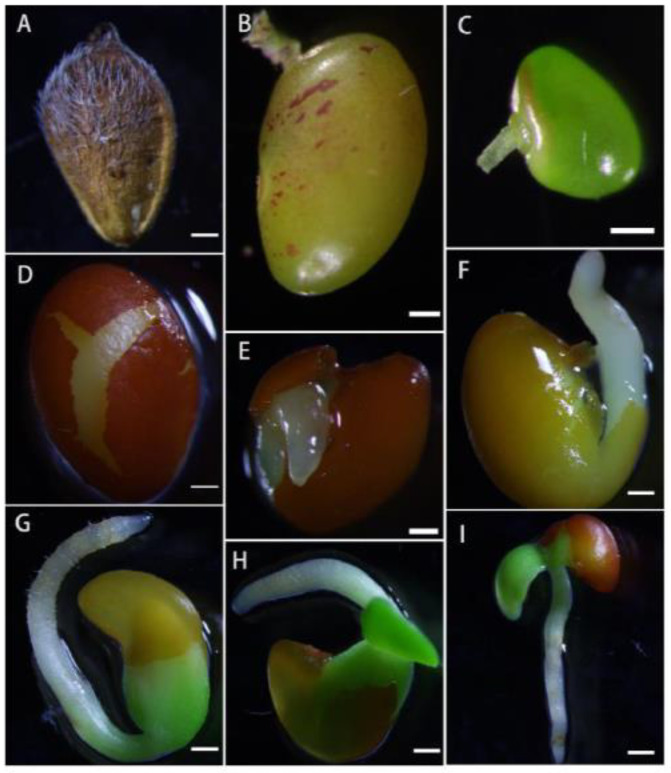
Microscopic observation of the seed structure in *L. davurica.* (**A**) Seeds of *L. davurica* are enclosed within the persistent calyx. Scale bar = 100 μm. (**B**,**C**) Observation of the seed hole and hilum. Scale bar = 100 μm. (**D**–**F**) The seed coat absorbs water, undergoes swelling, and subsequently, the radicle emerges by breaking through the seed coat. Scale bar = 100 μm. (**G**–**I**) Radicle, hypocotyl, and embryo. Scale bar = 100 μm.

**Figure 10 plants-13-01661-f010:**
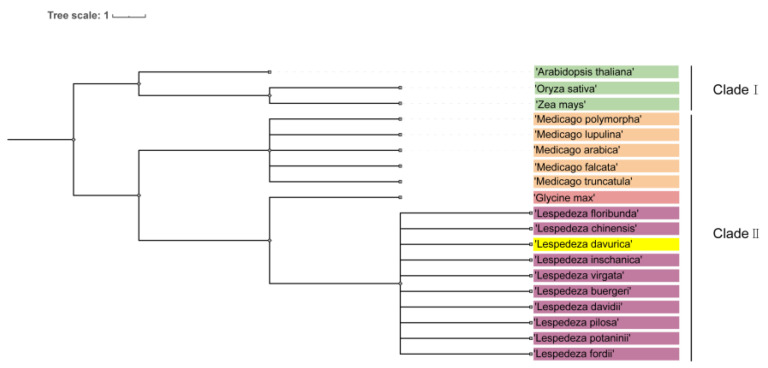
Phylogenetic tree analysis of *L. davurica*. The NCBI website was utilized to search for genomic information on *Arabidopsis thaliana*, *Oryza sativa*, *Zea mays*, *Lespedeza*, and *Medicago* genera, and an evolutionary tree was constructed using the common tree function available on NCBI.

## Data Availability

Data are contained within the article.

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
