# Peer review of "Morphological Study on the Differentiation of Flower Buds and the Embryological Stages of Male and Female Floral Organs in Lespedeza davurica (Laxm.) Schindl. cv. JinNong (Fabaceae)"

_plants, 2024, doi:10.3390/plants13121661_

Round 1

Reviewer 1 Report

Comments and Suggestions for Authors

Comments for the manuscript entitled "Morphological observation on flower bud differentiation and male and female flower organ structure and development of Lespedeza davurica cv. jinnong (Leguminosae)" submitted by Lirong Tong and Juan Wang.

The paper focusrs on the detailed description og the embryological chatscteristics of Lespedeza davurica cv. JinNong species. Also, tangentially reference is made about the expression of genes during floral organogenesis. During the study, clarifications are made on the models of development of the male gametophyte (pollen), of the female gametophyte (embryonic sac) of Lespedeza davurica

Details about double fertilization, zygote formation, stages of embryo development, formation and development of Lespedeza davurica endosperm are shown.

The strucrure and ripening process of Lespedeza davurica endosperm are shown.

The structure and ripening process of Lespedeza davurica seeds are also presented.

Finally, a phylogenetic tree analysis is presented, indicating that Lespedeza davurica belongs to the alfalfa and soybean branch.

This study is welcome because currently there are few studies on the reproductive biology of plants from  Leguminosae family.

The present study could also be practical importance in breeding programs to ensure seed yields in legume plant hybridization.

My comments are below:

1. The title could be reworded like this: Morphological study on the differentiation of flower buds and the embryological stages of male and female floral organs in Lespedeza davurica (Laxm.) Schindl. cv. JinNong (family Leguminosae). At the Keywords correctly is Leguminosae

2. In line 18 should In vivo.

3. In line 27 should: plants from the genera Lespedeza and Medicago.

4. In line 41 should Platycladus.

5. In line 44 should Koelreuteria elegans.

6. In line 48 should Magnoliaceae.

7. In line 66 should solitary flowers.

8. In lines 82-83 should: for the developmental biology and molecular biology of plants in  Leguminosae family and especially of  Lespedeza genus.

9. In line 313, under Figure 8, P and Q photos are missing, which you mention that represents "Formation of the germ in the mature embryo".

10. In line 359 should Leguminosae and Poaceae.

11. In line 362, instead of "Lespedeza and Alfalfa", correct is: species of Lespedeza and Medicago genera.

12. In line 450 you wrote that "Callose plays a crucial role in various stages of plant growth and development". Could you give a concrete example on this regard?

13. The abbreviations for ANJ and HERK1  should be accompanied by a parenthesis for the full name, as you have done with FER, RALF and RLK.

14. In lines 491-492 should Brassicaceae, Graminaceae, Leguminosae.

15. In lines 505-506 you mention that: "It is speculated that this asynchronous development of male and female pistils ..." There is no male pistils! Improve the sentence.

16. In line 512 you wrote: "In flowering plants two fertilization events occur ...". It is not correct! And gymnosperms are flower plants and have no double fertilization! It is correct: at angiosperms two fertilization events occur.

17. In lines 518-519 you should reformulate the phrase. That is, do not use the plural for sperm cell that fertilize the central nucleus of the embryonic sac! More correctly, it is to refer only to the two sperm cells produced by a pollen grain.

18. In lines 520-521, it is more correct that an embryo (not embryos) is formed from the zygote.

19. In line 552, should: alfalfa, soybean (not italic, without uppercase letters).

20. In line 555, should: alfalfa and clover.

21. In line 585, should: safranin stains, not Safranin stains!

22. In line 607, should: 4.5 Phylogenetic tree

I wish you much success in publishing this research article!

Reviewer 2 Report

Comments and Suggestions for Authors

The subject of research presented in the manuscript entitled “Morphological observation on flower bud differentiation and male and female flower organ structure and development of Lespedeza davurica cv. jinnong (Leguminosae)” written by Lirong Tong and Juan Wang is connected with the reproduction processes in plants. The plant species that was chosen by authors is known for its different properties and in some Asian countries is recommended in traditional medicine for treating diabetic patients.

The section “1. Introduction” contains too much general information about plant reproduction. It would be good to add more data or examples concerned the reproduction in Fabaceae family.

The second part of the paper (“2. Results”) is very large and contains numerous of results. This part contains 10 figures and among them 9 consist of (mostly) photographs which are arranged in a proper way. The large documentation in this sections is most often of good quality. And the description of the results obtained by authors is very detailed. The results of the research contain the effects of the analyses carried out using appropriate methods to the aims planned by the authors. When the figures are described, it would be proper to use e.g. …A,B… when you have only two photographs instead …A-B… . 

 The next section, "3. Discussion" is very large and is divided into 6 subsections which works well. Generally, this section is constructed in a typical manner for scientific publications. The results obtained by the authors are compared to the appropriate literature data. The text of the paper is supplemented with the conclusions, which are concise and properly formulated.

In the next, third part of the publication, entitled “4. Materials and Methods” the authors wrote about the materials and the methods which were used during performed study. They used appropriate techniques and describe them properly. I recommend using indexes, e.g. …with 1% K2HPO4 for…K2HPO4

Additional comments (only examples are given):

Line 4: Title of the paper – currently the authors use the name Fabaceae instead Leguminosae

Line 62: …perennial leguminous grass… check the mining of word “grass” –members of Fabaceae family are not grasses

Line: 125: use bold in name for Figure 1

Line 171: …ovule(Figure 3L-N)… should be ovule (Figure… - check spaces between the words and punctuation throughout the whole text

Line 667: repeated twice … 2010   – check the literature items carefully

To sum up, I am convinced that the reviewed paper can be published in “Plants,” but it needs minor revisions first.
